# Convolutional Neural Network-Based ECG-Assisted Diagnosis for Coal Workers

**DOI:** 10.3390/ijerph20010009

**Published:** 2022-12-20

**Authors:** Yujia Wang, Zhe Chen, Sen Tian, Shuxun Zhou, Xinbo Wang, Ling Xue, Jianhui Wu

**Affiliations:** 1Key Laboratory of Coal Mine Health and Safety of Hebei Province, School of Public Health, North China University of Science and Technology, No. 21 Bohai Avenue, Caofeidian New Town, Tangshan 063210, China; 2Jining Center for Disease Control and Prevention, No. 26 Yingcui Road, Rencheng District, Jining 272000, China; 3College of Science, North China University of Science and Technology, No. 21 Bohai Avenue, Caofeidian New Town, Tangshan 063210, China

**Keywords:** coal workers, ECG abnormalities, convolutional neural net, image recognition

## Abstract

Objective: To process and extract electrocardiogram (ECG, ECG, or EKG) features using a convolutional neural network (CNN) to establish an ECG-assisted diagnosis model. Methods: Coal workers who underwent physical examinations at Gequan Mine Hospital and Dongpang Mine Hospital of Hebei Jizhong Energy from July 2020 to September 2020 were selected as the study subjects. The ECG images were preprocessed. We use Python software and convolutional neural network to establish ECG images recognition and classification model.We usecalibration curve, calibration-in-the-large, Brier score, specificity, sensitivity, F1 score, Kappa value, accuracy, and area under the curve (AUC) of ROC to evaluate the performance of the model. Results: The number of abnormal ECG results was 849, and the rate of abnormal results was 25.02%. The test set accuracies of the sinus bradycardia model, nonspecific intraventricular conduction delay model, myocardial ischemia model, and sinus tachycardia model were 97.66%, 96.49%, 93.62%, and 93.02%, respectively; sensitivities were 96.63%, 96.30%, 96.88% and 95.24%, respectively; specificities were 98.78%, 96.67%, 86.67%, and 90.90%, respectively; Brier scores were 0.03, 0.07, 0.09, and 0.11, respectively; Calibration-in-the-large values were 0.026, 0.110, 0.041, and 0.098, respectively. Conclusions: The convolutional neural network model can accurately identify the main ECG abnormality types of coal workers. Additionally, the main ECG abnormalities in these coal company workers were sinus bradycardia, non-specific intraventricular conduction delay, myocardial ischemia, and sinus tachycardia.

## 1. Introduction

The lectrocardiogram, which detects electrophysiological signals from the heart muscle, is considered a favourable monitoring method [1]. It is used to assist in the diagnosis of cardiac diseases and plays an important role in determining the effects of drugs or electrolyte conditions on the heart and the status of artificial heart pacing [2]. The ECG is an essential and reliable diagnostic tool used in modern medicine [3,4,5]. However, due to the large number of physical examination personnel and the heavy workload, there are inevitably mistakes made in the recognition and diagnosis of electrocardiograms by doctors. Reliable automated interpretation of ECG signals is extremely beneficial for clinical routines and patient safety. When different doctors make diagnoses, the results are prone to errors. At the same time, manual recognition and diagnosis of ECGs by physicians is very time-consuming, and the long duration of interpretation work will produce fatigue, which will affect the physician’s judgment, thereby promoting misdiagnoses and affecting the accuracy [6].

The convolutional neural network is a kind of deep neural network commonly used in image recognition, which contains multiple hidden layers while introducing computational operations such as convolution and pooling [7] in the hidden layers [8,9,10] to reduce the dimensionality of the image, reduce the parameters of the network, and improve the training efficiency of the neural network, which is now widely used in the recognition and classification of various images, such as those from ECGs [11,12]. Additionally, the convolutional layer with overlapping repetitions in the front segment and pooling layers can perform deep extraction of image features and have better recognition and classification effects on images. With the development of technology, the real-time monitoring of dynamic ECGs based on machine learning has also developed rapidly [13,14,15], and the real-time state transmission and abnormality warning of dynamic ECGs promote more accurate and timely diagnoses. The standard database used by most researchers has obvious features and complete data, but in practical use, there are differences in ECG images detected by ECG machines with different leads or different parameters. After reviewing the data, we found that the ECG recognition models studied and adopted by many researchers have some limitations regarding the recognition of ECG abnormalities in different populations.

Hannun A Y [16] et al.’s machine learning group used a 34-layer convolutional neural network to detect arrhythmias. Compared with experts, the AUC value of their model reached 0.97, and the sensitivity was 0.83, which was higher than the expert level of 0.78.

Another scholar used CNN to identify the heartbeat and the input-segmented heartbeat, then used the knowledge of CNN to obtain features, and then used more perceptrons to recognize and analyze the input features, which is an end-to-end learning network structure [17,18]. Acharya [19] et al. used CNNC for the recognition of five heartbeats and inputted the heartbeat data after segmentation; their recognition accuracy reached 94.04% and the model performed well. Shun-Chi Wu et al. [20] proposed a deep CNN-based ECG signal recognition method, where deep convolutional neural networks could better extract ECG features. Thus, the recognition effect of this model was better. Ke Zhou [21] et al. proposed a support vector machine-based ECG classification and diagnosis method in 2006. Some methods combine multiple mathematical models for ECG image recognition, such as the combination of SVM and the CNN for recognition [22].

This study intends to establish an ECG-assisted diagnostic model using convolutional neural networks for the common types of ECG abnormalities in coal workers to achieve an efficient and accurate ECG diagnosis for the medical examination of occupational workers. The purpose of this study is to help physicians to perform ECG differentiation, reduce the workload of ECG diagnosis for physicians, and control the errors of manual diagnosis, which provide reference for the ECG diagnosis of the occupational population and contribute to the prevention and treatment of cardiovascular diseases. The purpose of this study is to assist doctors in electrocardiogram identification, reduce the workload of electrocardiogram diagnosis, and control the error rate of manual diagnosis. It provides a reference for the physical examination electrocardiogram diagnosis of the occupational population and also provides help for the prevention and treatment of cardiovascular diseases.

## 2. Materials and Methods

### 2.1. Study Subjects

The current situation study method was used in this study, and coal workers who attended occupational and health checkups at Dongpang Mine Hospital and Gequan Mine Hospital of Jizhong Energy Xing Coal Group from July 2020 to September 2020 were selected as the study population. A total of 3392 coal workers were eventually included in this study, and the content of this study was approved by the ethics committee of the North China University of Technology. Inclusion criteria: age between 18 and 60 years and signed informed consent; one year or more of service. Exclusion criteria: those with incomplete general information and electrocardiogram examination data.

### 2.2. Data Collection

First, general information about the subject was collected, including gender, age, education level, and marital status. Second, the examination was performed using a Biocare ECG-1210 twelve-lead digital ECG machine by a cardiologist with more than ten years of experience.

### 2.3. ECG Signal Pre-Processing

The ECG is heartbeat sliced and whitened by Python software [23], and later, the intercepted ECG targets are scaled and integrated into the same length using image downsampling for recognition and classification using CNN.

#### 2.3.1. Heartbeat Cut Score

In this study, the ECG data were preprocessed, and the ECG signal was finally segmented according to the heartbeat by a series of processes, such as QRS wave detection and grouping, selection of the main heartbeat, discrimination of premature beats, and detection of P-wave F-wave [24]. Then, the ECG image was sliced by heartbeat (a heartbeat is a bioelectric signal generated by a whole cardiac cycle) using convolutional neural network. A continuous P-wave, QRS wave cluster, and T-wave constitute a complete heartbeat, see Figure 1.

#### 2.3.2. Whitening Treatment

Whitening removes the correlation between the pixel points within the ECG signal so that the average pixel value of the image is 0 and the variance of the image is unit variance 1. Whitening the ECG signal feature segment image data reduces the redundancy of the data input.

(1)Zero-meaning

When data mining is performed, if the data is analyzed by direct input, the recognition may be affected by the small difference between the data compared to the absolute value. Therefore, the mean value of the signal should be zero first.

(2)Whitening transformation

The data after zero-meaning is whitened and transformed so that the new vector covariance matrix is a unit matrix, making the variance of the image a unit, variance 1.

In this study, the most common red, green, and blue (RGB) three channels are used. Each element of the input data are three-dimensional data, and when the three channels are equal (i.e., 255, 255, 255 grayscale map in this study), the element is preserved. As shown in Figure 2.

#### 2.3.3. Image Downsampling

Using image downsampling to adjust all images to the same scale, the final scale of the image in this study is adjusted to 256 × 256, making the image conform to the size of the input display area of the convolutional neural network. s-fold downsampling is performed on an image, and the image resolution is calculated as follows:(1)$$p_{k}=\sum_{i\epsilon win(k)}I_{i}/S^{2}$$

In the formula:
$p_{k}$—Image pixel average,I—Original image size,S—Downsampling multiplier.

### 2.4. Building the Mod

A convolutional neural network model was constructed using the Sklearn (SciKit-Learn). The four categories of ECG were divided into 10 random data sets in turn, with a training set: test set: and validation set ≈ 7:2:1. The training set was trained, and the average value of all operation results was used as the evaluation index of the model. The training set was used to train the model many times, and then the model coefficients were used to identify and classify the test set. Validation sets were used to verify the model performance, and the model performance was evaluated according to the degree of differentiation and calibration.

Using the corresponding Sequential functions to build CNN models, many deep learning frameworks have been developed into interfaces for each layer, and the deep model language expressiveness becomes more advanced, thereby becoming able to solve various intricate classification problems and making it easier to train [25]. The convolutional layer is a feature extractor that automatically extracts deep information from the input signal; the convolutional layer generates multiple feature maps based on the input information, and the pooling layer samples the generated feature maps to eliminate the redundant amount of data while retaining the useful data [26]. In this study, two-dimensional convolution is used, and the number of convolutional, pooling, and fully connected layers included in the network structure is 3. The convolutional and pooling layers are repeatedly and alternately stacked at the front end of the network to extract the features of the input image, while relu [27] is selected as the activation function for error reduction, and each convolutional layer is followed by the activation function relu to achieve maximum pooling, followed by the fully connected layer (flatten layer), which contains 512 neurons and use dropout 0.5 to adjust the parameters to prevent overfitting of the model. Two more fully connected layers (sense layers) are added, and finally, the dropout layer is added.

The application of deep convolutional neural networks in visual tasks such as object detection and image classification has become a mainstream method in the current field of computer vision [28]. The convolutional kernel scans the image to obtain the output data of the feature map (feature map). The convolutional layer in the front part of the network structure is used to extract the detailed features of the image, the extracted information is outputted as an image, and each of its pixels is a small part of the feature information of the input image, after which, each layer of convolutional layer for feature extraction gradually increases the range, enabling the discovery of deeper feature information from the image. Through the operation of multilayer convolution, the abstract representation of the image in different scales is finally obtained, as shown in Figure 3.

After the pooling layer, which reduces the dimension of the convolution layer, the maximum pooling is selected in this study, and the maximum value in the limited sampling range in the pooling window is used as the output value of the range. In this paper, a 2 × 2 matrix of size is selected, the window slides 2 steps at a time, and the input image of the pooling layer is split into different regions, after which, each element of the output is the maximum value of the elements in the divided region, as shown in Figure 4.

The output size of the convolution layer is obtained by the following equation:(2)$$\;n=\left(m-f+1\right)/s$$

The output size of the pooling layer is obtained by the following equation:(3) n=(m−f)/s+1

In the two equations:
n—Output image edge length,m—Input image edge length,f—Convolution kernel length,s—Convolution kernel sliding step.

### 2.5. Statistical Methods

We used SPSS 22.0 statistical analysis software from IBM USA to plot the ROC curves, AUC values were calculated, and differentiation evaluation metrics such as sensitivity, specificity, kappa values, and F1 score of the model were calculated using the cross-classification process of cross-tabulation. Python software was used for image pre-processing, construction of the convolutional neural network ECG recognition model, and calculation of calibration degree evaluation indexes such as Brier score, Calibration-in-the-large [29], and calibration curve of the model [30].

## 3. Results

### 3.1. Basic Information

A total of 3392 respondents were included in the study, including 3268 males (96.34%) and 124 females (3.66%); the average age was 39 (34, 47) years; 1418 (41.81%) were in junior high school and below, and 3248 (95.78%) were married. We collected and collated the basic data of coal workers. As shown in Table 1.

### 3.2. Detection of ECG Abnormalities

The ECG results of each worker were separately diagnosed by three cardiologists with associate senior titles. If the results of the three people are consistent, the diagnosis result is the same. If the results of the three people are inconsistent, the majority rule is adopted. If the results of the three people are inconsistent, the consultation of the three people is carried out. The result of the consultation is the diagnostic result and is the gold standard of electrocardiogram.
(4)$$Time\sim O\left(\sum_{l=1}^{D}M_{l}^{2}K_{l}^{2}C_{l-1}C_{l}\right)$$

*D*: the depth of the network.*l*: the *l*th convolution layer of the neural network.*C_l_*: number of convolutional kernels in this layer.

For the L-th convolution layer, the number of input channels $C_{in}$ is the number of output channels for the L-1st convolution laye-r. The time complexity of this model is:$$256^{2}\cdot 3^{2}\cdot 3\cdot 32+256^{2}\cdot 3^{2}\cdot 3\cdot 64+256^{2}\cdot 3^{2}\cdot 3\cdot 128=3.96361728\times 10^{8}$$

There were 849 abnormal ECG results, with an abnormality rate of 25.02%, of which 838 (98.70%) were of one abnormal type and 11 (1.30%) were of two abnormal types. The main abnormal types were sinus bradycardia, nonspecific intraventricular conduction delay, myocardial ischemia, and sinus tachycardia, accounting for 51.94%, 17.31%, 13.90% and 8.59% of the total number of abnormalities, respectively, as shown in Table 2.

### 3.3. ECG Image Processing

The region selection of ECG image results was performed using a convolutional neural network to intercept the II-lead ECG images of consecutive heartbeats, and some of the original heartbeats intercepted in this paper are shown, including four typical abnormal categories and normal sinus rhythm images, as shown in Figure 5a,b.

### 3.4. Results of Image Recognition Models for the Four Main Anomaly Types

The four main abnormality types were randomly divided into the training set, test set, and validation set according to the ratio of 7:2:1 with normal ECG, respectively, and the corresponding convolutional neural network abnormality image recognition model was established. The classification results of the data set were obtained and the model prediction results were compared with the actual results of the samples to obtain the confusion matrix.

#### 3.4.1. Sinus Bradycardia Image Recognition Model

In its training set, six hundred fifty-four samples were correctly classified and eight samples were incorrectly classified, with an accuracy rate of 98.79%; in the test set, one hundred sixty-seven samples were correctly classified and four samples were incorrectly classified, with an accuracy rate of 97.66%; in the validation set, ninety-four samples were correctly classified and three samples were incorrectly classified, with an accuracy rate of 96.91%, as shown in Table 3.

#### 3.4.2. Non-Specific Indoor Conduction Delay Image Recognition Model

Its training set had two hundred and nine samples correctly classified and four samples incorrectly classified, with an accuracy of 98.12%; the test set had fifty-five samples correctly classified and two samples incorrectly classified, with an accuracy of 96.49%; the validation set had twenty-seven samples correctly classified and two samples incorrectly classified, with an accuracy of 93.10%, as shown in Table 4.

The sensitivity of the training set is 98.17% and the specificity is 98.08%; the sensitivity of the test set is 96.30% and the specificity is 96.67%; the sensitivity of the validation set is 93.75% and the specificity is 92.31%.

#### 3.4.3. Myocardial Ischemia Image Recognition Model

Its training set had one hundred thirty-nine samples correctly classified and four samples incorrectly classified, with an accuracy of 97.20%; the test set had forty-four samples correctly classified and three samples incorrectly classified, with an accuracy of 93.62%; the validation set had twenty-one samples correctly classified and two samples incorrectly classified, with an accuracy of 91.30%, as shown in Table 5.

The sensitivity of the training set was 97.10% and the specificity was 97.30%; the sensitivity of the test set was 86.67% and the specificity was 96.88%; the sensitivity of the validation set was 90.90% and the specificity was 83.33%.

#### 3.4.4. Sinus Tachycardia Image Recognition Model

Its training set had ninety-nine samples correctly classified and four samples incorrectly classified, with an accuracy rate of 96.12%; the test set had forty samples correctly classified and three samples incorrectly classified, with an accuracy rate of 93.02%; the validation set had twenty-one samples correctly classified and two samples incorrectly classified, with an accuracy rate of 91.30%, as shown in Table 6.

The sensitivity of the training set was 95.16% and the specificity was 97.56%; the sensitivity of the test set was 95.24% and the specificity was 90.90%; the sensitivity of the validation set was 92.31% and the specificity was 90.00%.

### 3.5. ECG Recognition Model Performance Evaluation

The training set, test set, and validation set of model samples are divided into groups, and the effect of four abnormal types of ECG image recognition models is evaluated with the index of differentiation. Calibration curves of Brier, F1 Score, and test set results were used to evaluate the degree of calibration.

#### 3.5.1. Training Set Model Effect Evaluation

The F1 scores for the training set of the model in identifying four categories of ECG images, namely, sinus bradycardia, nonspecific intraventricular conduction delay, myocardial ischemia, and sinus tachycardia, were 0.99, 0.98, 0.97, and 0.96, respectively; the kappa values were 0.98, 0.96, 0.94, and 0.92, respectively; the AUCs were 0.988, 0.981, 0.972, and 0.947, respectively, as shown in Table 7, and the ROC curves are shown in Figure 6a–d.

#### 3.5.2. Test Set Model Effect Evaluation

The F1 scores for the test set of the model in identifying four categories of ECG images, sinus bradycardia, non-specific intraventricular conduction delay, myocardial ischemia, and sinus tachycardia, were 0.98, 0.97, 0.92 and 0.93, respectively; Kappa values were 0.95, 0.93, 0.85 and 0.88, respectively; AUC was 0.977, 0.920, 0.869 and 0.894; Brier scores were 0.03, 0.07, 0.09, and 0.11; Calibration-in-the-large values were 0.026, 0.110, 0.041, and 0.098, respectively. See Table 8, calibration curves are shown in Figure 7a–d, and ROC curves are shown in Figure 8a–d.

#### 3.5.3. Validation Set Model Effect Evaluation

The F1 scores for the validation set of the model in identifying four categories of ECG images: sinus bradycardia, nonspecific intraventricular conduction delay, myocardial ischemia, and sinus tachycardia were 0.97, 0.93, 0.87, and 0.91, respectively; the kappa values were 0.94, 0.86, 0.83, and 0.82, respectively; and the AUCs were 0.970, 0.930, 0.931, and 0.912, respectively, as shown in Table 9. The ROC curves are shown in Figure 9a–d.

## 4. Discussion

Electrocardiography is currently the most common and convenient method for examining cardiac rhythm and myocardial blood supply or conduction and diagnosing cardiac health. It is widely used in the diagnosis of cardiovascular diseases and arrhythmias [31]. Performing manual ECG diagnosis is very time-consuming. When physicians must continuously observe waveforms over a long period of work, they become fatigued, and cannot easily identify ECG abnormalities, which can lead to misdiagnosis and missed diagnoses. In recent years, machine learning has been gradually applied to the field of ECG diagnosis [32,33]. As a branch of machine learning, deep learning [34], has achieved great success in determining abnormal ECGs and ECG waveforms, which greatly improving the accuracy of diagnosis. Fatma Murat [35] et al. reviewed 24 relevant articles published in international journals and found that most of the studies used the CNN model and achieved good results.

ECG examination has become one of the most important examination items in health checkups. The functional status of the heart can be judged by analyzing ECG results. A total of 3392 coal workers were included in this study, and the results showed that the number of ECG abnormalities was 849, with an abnormality rate of 25.02%, which mainly included sinus bradycardia, nonspecific intraventricular conduction delay, myocardial ischemia, and sinus tachycardia, accounting for 90.46% of the total abnormalities. The results of the study were similar to the results of routine ECG examinations performed by Wu Wei [36] on 4112 coal miners. The long-term overload and high-intensity work of coal miners can lead to increased cardiac load, causjng abnormal cardiac function, and triggering the development of cardiovascular diseases. An epidemiological survey study [37] showed that the more years a coal worker spent working underground, the higher the incidence of a series of cardiovascular diseases, such as coronary heart disease and hypertension. Adam Goldman [38] et al. found that ECG abnormalities were associated with various pathological processes and could be the first sign of underlying cardiac diseases by studying and analyzing the ECGs of 2769 men and women. In the ECG analysis of coal miners, research by Li Rongrun [39] shows that the types of abnormal ECGs with high incidence rates were sinus arrhythmia or sinus bradycardia (31.32%) and limb conduction hyper voltage (25.46%). Sinus rhythm abnormalities in coal workers are the result of prolonged physical labor, and pathogenic causative factors are rarely present. In the analysis of the ECG results of more than 40,000 physically examined workers in our hospital, C.C. Liu [40] found ST-T segment abnormalities at the highest rate, followed by sinus bradycardia, sinus tachycardia, and bundle branch block, which was similar to the results of the present study. Similarly, in the study by Xiaoqin Gao [41], the rate of ST-T segment abnormalities was also found to be higher in the healthy population, which was similar to the results of the present analysis. The above research results shows that ST-T segment alterations (suggestive of myocardial ischemia) are a relatively common ECG abnormality, and their abnormalities are associated with hypertension and primary cardiomyopathy. Further investigations are recommended for this population. Li Yi [42] et al. conducted a twelve-lead routine ECG examination on 9807 healthy male youths enlisted in a ministry of the armed forces, and the results of the study showed that sinus arrhythmia and sinus bradycardia were both exhibited at a high rate in this population, similar to the results detected in the present study, as a result of high-intensity exercise or high-intensity work, making them prone to sinus bradycardia. The detection of abnormal ECG and the detection rate vary in different populations due to the different working environments and working hours. Cheng Shuyan [43] et al. examined the ECG of 3000 labor dispatchers abroad and found that the detection rate of abnormal ECG in this population was 4.27%. Among them, the results of the analysis of abnormal ECGs showed that myocardial ischemia accounted for a higher percentage of abnormal ECGs, which was similar to the findings of our study. Xiao Yanrong [44] et al. performed a twelve-lead routine ECG examination on 3626 college students and statistically analyzed the abnormal ECGs in this population. The results of the study showed that sinus bradycardia, sinus tachycardia, and sinus arrhythmia were the most common among the abnormal ECGs. Wu Yiqin [45] obtained a noise OR value of 1.459 (1.108~1.923) in the analysis of the health status of workers in a textile printing and dyeing enterprise in Yixing, indicating that noise is an independent factor affecting ECG abnormality, and the detection rate of ECG abnormalities in workers exposed to noise was 45.9% higher than that of workers without noise exposure. The types of ECG abnormalities and their ratios vary among different populations, but coal workers are at higher risk of ECG abnormalities and cardiovascular diseases due to their high work intensity, frequent shift work, and the presence of various occupational harmful factors such as dust, noise, and high temperatures in the working environment.

Neural networks can be used for the classification and prediction of various data, and convolutional neural networks for recognition of ECGs need enough training data, i.e., they should contain enough experience for the model to learn to get a good training effect. If the data are not balanced, it will affect the model’s effectiveness [46,47]. In this study, when we used the multi-category diagnosis model in the early stage, the large difference in the number of samples for each disease category led to poor recognition performance in the final model. Based on this, in this study, the recognition models of convolutional neural networks were constructed by identifying the main ECG abnormality types of four coal workers, and each type of disease was performed separately to equalize the positive and negative sample ratios with a similar number of normal sinus rhythm ECGs to ensure the recognition effect of the models. Through the four models finally obtained from the study, it can be found that the sinus bradycardia model has the best classification recognition effect, and the accuracy rates of all three sample sets are above 95%, from which it can also be concluded that the larger the number of samples, the higher the accuracy rate of the model, and the better the recognition effect. Shenhua Liu [48] et al. classified and recognized 34 types of abnormal ECGs based on a multi-convolutional ResNet network structure with 20,000 ECG data, and the model results showed an F1 score value of 0.91, and accuracy and recall rates of 93.96% and 87.89%, respectively, which also confirmed this view. Cui Jianfeng [49] et al. used the MIT-BIH [14,15,50] arrhythmia database for model training and applied all 47 data from this library, and their results showed an accuracy of 99.00% and a recall of 99.08%. Yıldırım Ö [51] et al. used CNN to conduct experiments using the MIT-BIH database to ensure the quality of the images and feature distinctness and proposed a method for ECG signal recognition based on long-time ECG, which performed better according to the results of this study.

Convolutional neural networks are more sensitive to the waveforms of ECG images and are, therefore, more often used in the field of image recognition, especially ECG classification and recognition. In the study by Wang Eric Ke [52], a convolutional neural network was used with a bidirectional recurrent neural network model, and its accuracy was found to be 87.69% in a model validation of 120,000 ECG data samples. Wang Guanjun [53] et al. used a one-dimensional convolutional neural network to identify more than 20,000 pieces of myocardial infarction ECG data, and its test set accuracy was 0.952; some scholars also used different deep learning methods for ECG identification. The results of the study showed that the identification accuracy of the convolutional neural network was higher [33]. The effectiveness of convolutional neural network models for ECG recognition not only depends on the number of images (i.e., sample size) but also on the different methods of model building. Different construction methods can have an impact on the effect of convolutional neural network ECG recognition. The convolutional neural network model with four, three, and two convolutional, pooling, and fully connected layer structures proposed by Yannan Wang [54] et al. and the multi-kernel multi-scale convolutional neural network classification model proposed by Yan Wu [55] have better results than single-kernel convolutional and other deep learning models, and the number of convolutional layers and methods used to build the model in the above two groups of studies are different from this study, but the model affects performance in this study similar to the above studies. This indicates that the selection of the model-building method in this study is more reasonable, while the recognition accuracy in the ECG recognition model based on convolutional neural networks conducted by some researchers [56,57] is slightly lower, which may be related to the model structure and some parameters chosen by them. Liu W [58] et al. used MFB-CBRNN to conduct a five-category recognition study of 12-lead ECG, and its overall accuracy was 93.08%. This model can simultaneously perform the recognition of five categories of ECG. This study modeled different kinds of ECG abnormalities separately, and the results showed that each model has better recognition and higher accuracy for ECG images, and the subsequent model collection approach can be used to establish a set of ECG recognition models to improve the ease of use. Romdhane TF [59] et al. used a more complex method for the ECG recognition study, and the accuracy of recognition is similar to our study. In conclusion, the ECG recognition model developed in this study has a high accuracy rate and can be used as an auxiliary diagnostic tool for ECG examination in the health examination of coal workers.

The convolutional neural network model established in this study is effective for the recognition of ECG of coal workers, but the sample size of the selected population is relatively small, and the number of some ECG abnormality types is insufficient to achieve recognition and classification of this part of ECG abnormalities at this stage, and the sample size should be further expanded in subsequent studies to establish recognition models for various ECG abnormalities, The recognition models were established for each ECG abnormality category, and further model sets should be established to improve the convenience of use. When the ECG classification of normal and four arrhythmia diseases were performed based on the MIT-BIH database, the accuracy rate reached 98.82%. Compared with the existing classification methods, the algorithm in this paper improves the accuracy of automatic ECG classification with good generalization ability. However, due to the limitation of data collection, retired workers over 60 years old were not included in the study, which may lead to the deviation of the research results. Follow-up studies can be conducted on retired workers to assess the impact of age on the bone mineral density of underground coal mine workers.

## 5. Conclusions

Sinus bradycardia, sinus tachycardia, nonspecific intraventricular conduction delay, and myocardial ischemia were the top four main abnormal ECG findings in the subjects of this study. The constructed convolutional neural network model has good discrimination and calibration for ECG image recognition and diagnosis, which can help doctors make aided diagnoses more accurately.

## Figures and Tables

**Figure 1 ijerph-20-00009-f001:**
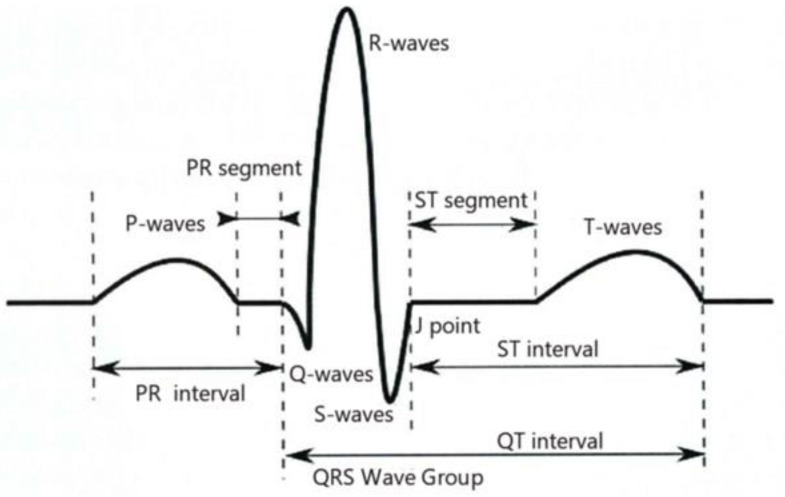
Schematic diagram of heartbeat.

**Figure 2 ijerph-20-00009-f002:**
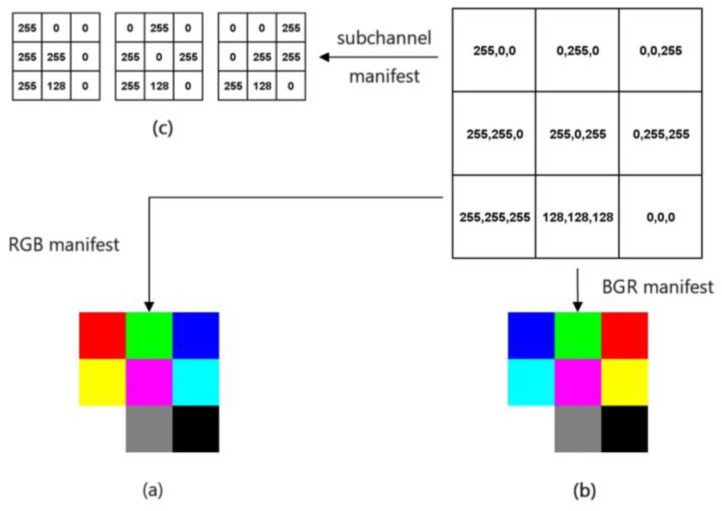
Schematic diagram of the whitening process of ECG signal: (**a**,**b**) Red, green and blue (RGB) three-channel diagram; (**c**) Sub-channel representation results.

**Figure 3 ijerph-20-00009-f003:**
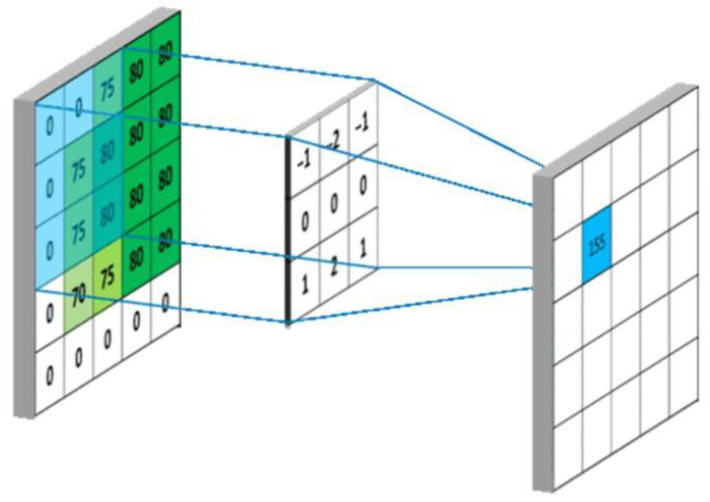
Convolutional layer output diagram.

**Figure 4 ijerph-20-00009-f004:**
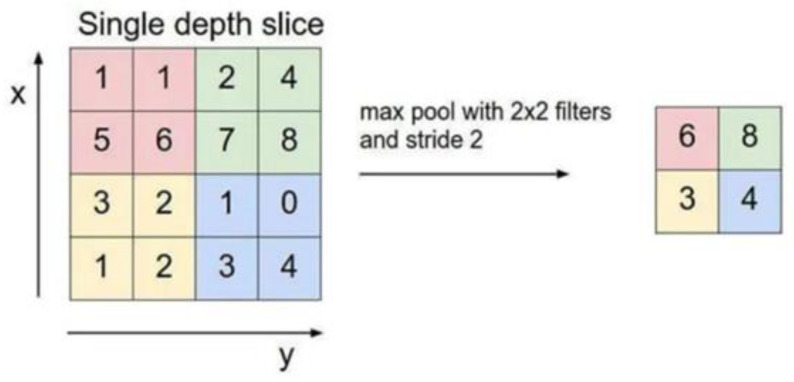
Schematic diagram of pooling layer output.

**Figure 5 ijerph-20-00009-f005:**
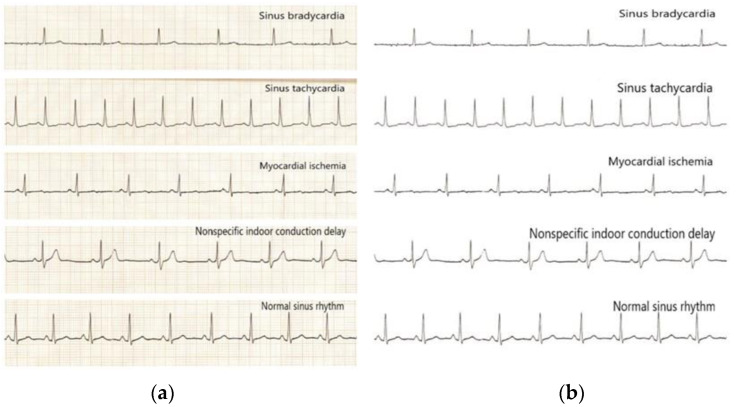
Images of the four typical abnormal categories and normal sinus rhythm before and after processing: (**a**) Before processing; (**b**) After processing.

**Figure 6 ijerph-20-00009-f006:**
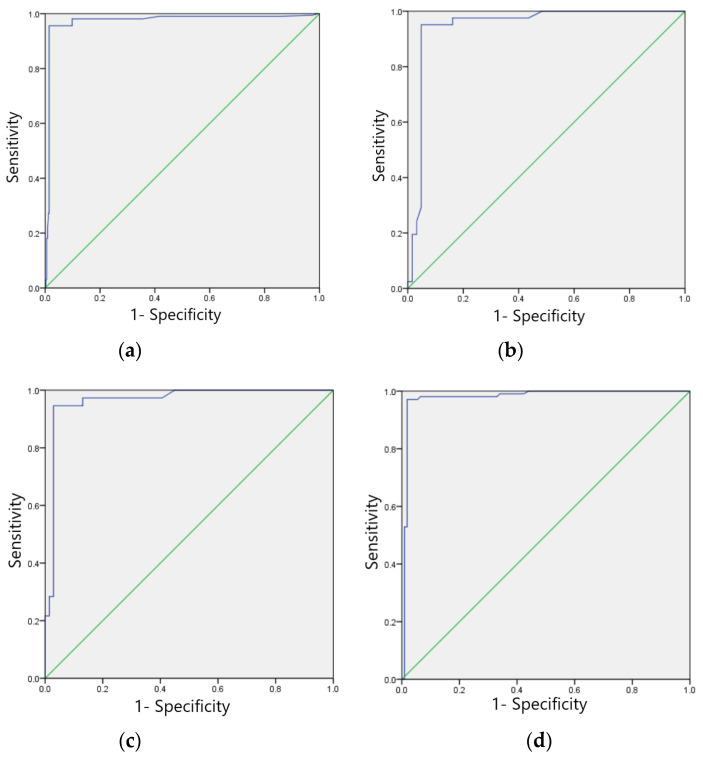
ROC curves of training sets of four typical anomaly categories: (**a**) Sinus bradycardia; (**b**) Sinus tachycardia; (**c**) Myocardial ischemia; (**d**) Nonspecific intraventricular conduction delay.

**Figure 7 ijerph-20-00009-f007:**
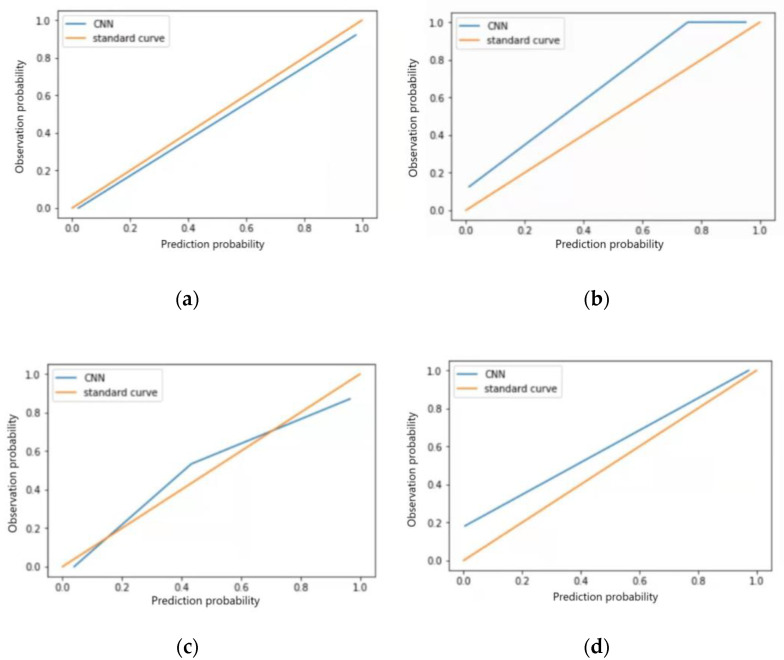
Calibration curves for four typical anomaly categories of test sets: (**a**) Sinus bradycardia; (**b**) Sinus tachycardia; (**c**) Myocardial ischemia; (**d**) Nonspecific intraventricular conduction delay.

**Figure 8 ijerph-20-00009-f008:**
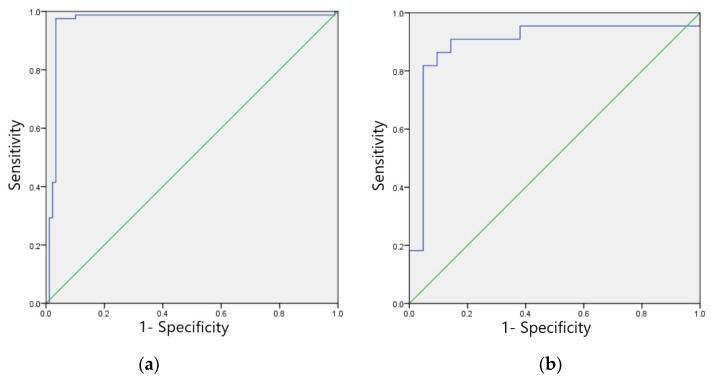
ROC curves for four typical anomaly categories of test sets: (**a**) Sinus bradycardia; (**b**) Sinus tachycardia; (**c**) Myocardial ischemia; (**d**) Nonspecific intraventricular conduction delay.

**Figure 9 ijerph-20-00009-f009:**
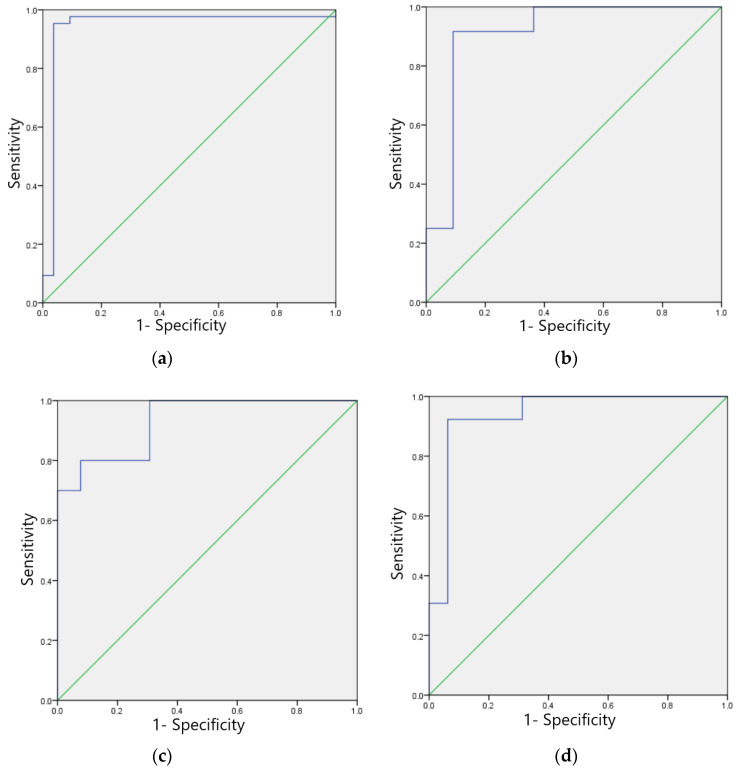
ROC curves of the verification set of four typical anomaly categories: (**a**) Sinus bradycardia; (**b**) Sinus tachycardia; (**c**) Myocardial ischemia; (**d**) Nonspecific intraventricular conduction delay.

**Table 1 ijerph-20-00009-t001:** Basic information of coal workers.

Basic Information	Group	Quantity	Percentage (%)
Gender	Man	3268	96.34
	Woman	124	3.66
Age	<30	197	5.81
	30~	1573	46.37
	40~	1030	30.37
	≥50	592	17.45
Degree of education	Junior High School and below	1418	41.81
	High school	1096	32.31
	College degree or above	878	25.88
Marital status	Unmarried	106	3.13
	Married	3249	95.78
	Other	37	1.09

**Table 2 ijerph-20-00009-t002:** Detection of abnormal ECG in coal workers.

Exception Category	Number of People	Percentage (%)
Sinus bradycardia	441	51.94
Non-specific intraventricular conduction delay	147	17.31
Myocardial ischemia	118	13.90
Sinus tachycardia	73	8.60
Left ventricular hypertrophy	24	2.83
Premature ventricular contractions	14	1.65
Supraventricular premature contractions	10	1.18
Ectopic atrial rhythm	9	1.06
Atrioventricular block	7	0.82
WPW syndrome (type B)	6	0.71
Hypertrophy of the right ventricle	5	0.59
Ectopic premature contractions	3	0.35
Intersectional tachycardia	1	0.12
Atrial fibrillation	1	0.12
WPW syndrome (type A)	1	0.12

**Table 3 ijerph-20-00009-t003:** Classification results of sinus bradycardia image recognition model.

Zoning	Correct	Error	Accuracy (%)
Training sets	654	8	98.79
Test Sets	167	4	97.66
Validation Sets	94	3	96.91

**Table 4 ijerph-20-00009-t004:** Classification results of non-specific intraventricular conduction delay image recognition model.

Zoning	Correct	Error	Accuracy (%)
Training sets	209	4	98.12
Test Sets	55	2	96.49
Validation Sets	27	2	93.10

**Table 5 ijerph-20-00009-t005:** Classification results of myocardial ischemia image recognition model.

Zoning	Correct	Error	Accuracy (%)
Training sets	139	4	97.20
Test Sets	44	3	93.62
Validation Sets	21	2	91.30

**Table 6 ijerph-20-00009-t006:** Classification results of sinus tachycardia image recognition model.

Zoning	Correct	Error	Accuracy (%)
Training sets	99	4	96.12
Test Sets	40	3	93.02
Validation Sets	21	2	91.30

**Table 7 ijerph-20-00009-t007:** Evaluation of the effect of ECG category recognition training set.

Types of ECG Abnormalities	Sensitivity (%)	Specificity (%)	F1 Score	Accuracy (%)	Kappa Value	AUC	PPV (%)	NPV (%)
Sinus bradycardia	98.55	99.05	0.99	98.79	0.98	0.988 (0.978~0.998)	99.13	98.44
Non-specific intraventricular conduction delay	98.17	98.08	0.98	98.12	0.96	0.981 (0.960~0.998)	97.61	98.53
Myocardial ischemia	97.10	97.30	0.97	97.20	0.94	0.972 (0.941~0.989)	97.30	97.11
Sinus tachycardia	95.16	97.56	0.96	96.12	0.92	0.947 (0.897~0.996)	98.36	93.07

**Table 8 ijerph-20-00009-t008:** Evaluation of the effect of the ECG category recognition test set.

Types of ECG Abnormalities	Sensitivity (%)	Specificity (%)	F1 Score	Accuracy (%)	Kappa Value	AUC	Brier Score	Calibration-in-the-Large	PPV (%)	NPV (%)
Sinus bradycardia	96.63	98.78	0.98	97.66	0.95	0.977 (0.951~0.995)	0.03	0.026	99.85	96.42
Non-specific intraventricular conduction delay	96.30	96.67	0.97	96.49	0.93	0.920 (0.835~0.993)	0.07	0.110	96.48	96.50
Myocardial ischemia	86.67	96.88	0.92	93.62	0.85	0.869 (0.810~0.987)	0.09	0.041	92.87	93.94
Sinus tachycardia	95.24	90.90	0.93	93.02	0.88	0.894 (0.842~0.988)	0.11	0.098	90.90	95.24

**Table 9 ijerph-20-00009-t009:** Evaluation of the effect of ECG category identification validation set.

Types of ECG Abnormalities	Sensitivity (%)	Specificity (%)	F1 Score	Accuracy (%)	Kappa Value	AUC	PPV (%)	NPV (%)
Sinus bradycardia	96.30	97.67	0.97	96.91	0.94	0.970 (0.931~0.992)	97.77	95.49
Non-specific intraventricular conduction delay	93.75	92.31	0.93	93.10	0.86	0.930 (0.820~0.972)	93.37	92.40
Myocardial ischemia	90.90	83.33	0.87	91.30	0.83	0.931 (0.777~0.958)	87.80	99.20
Sinus tachycardia	92.31	90.00	0.91	91.30	0.82	0.912 (0.772~0.949)	91.74	90.09

## Data Availability

Data available on request due to restrictions privacy. The data presented in this study are available on request from the corresponding author. The data are not publicly available due to the data are not readily available.

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
