# Peer review of "Convolutional Neural Network-Based ECG-Assisted Diagnosis for Coal Workers"

_ijerph, 2022, doi:10.3390/ijerph20010009_

Round 1

Reviewer 1 Report

I am very honored to be able to participate in the review of this paper. This paper focused on an ECG-assisted diagnosis model using a convolutional neural network (CNN) for the procession and extraction of the electrocardiogram (ECG, ECG, or EKG) features. In general, I think that this is both an interesting and valuable paper that describe a diagnostic application for the common types of ECG abnormalities in coal workers. There are some questions that the authors should answer.

1. In the image reduction sampling module an account should be given of what is the optimal size of the input display area for the convolutional neural network.

2. In the basic information module of the results, a brief analysis of the main components of the surveyed population and its impact on the experiment should be provided.

3. In the last sentence in 3.5.3, whether the initial letter is capitalized or not, and in the last sentence in 3.5.2, whether the initial letter is capitalized or not; and, the icons in Figure 2 are inconsistent with other icons in text, and the sizes of the four figures (a), (b), (c), and (d) in Figure 8 are inconsistent with the sizes of the four figures (a), (b), (c), and (d) in Figure 9

4. The materials and methods do not explain how to perform the model evaluation.

5. The F1 scores mentioned in the table of evaluation indicators are not covered in the 2.5 statistical methods.

6. How many doctors diagnosed the ECG physical examination results? If multiple, how to deal with different diagnostic results among doctors.

7. Some parts of the discussion are repeated with the introduction.

8. Reference 23 "Research on automatic recognition algorithm of static ECG based on machine learning" is marked above "python software" in the paper, 24 "Application of Python language in big data analysis "The two are reversed.

9. The higher incidence of disease in individuals over 60 years of age may be a selection bias. Can the authors explain the solution to this problem?

10. The research results of "Ding Hua" cited in the discussion section deviate greatly from the content of the present study, should this citation be revised.

Author Response

Point 1:  In the image reduction sampling module an account should be given of what is the optimal size of the input display area for the convolutional neural network.

Response 1:Thank you for reminding me. The image scale used for sampling is 256×256, which is in line with the size of the convolutional neural network input display area. Relevant content of this part has been mentioned in line 164 of subheading 2.3.3. For some reason, you may not have noticed this.

Point 2: In the basic information module of the results, a brief analysis of the main components of the surveyed population and its impact on the experiment should be provided.

Response 2:Thank you for your professional advice. We have supplemented the general information of the subjects, as shown in Table 1 of the subheading 3.1 Basic Information of Coal Workers. In the module 3.2 Abnormal Detection of Electrocardiogram, we described and listed the abnormal electrocardiogram of the study object, which we believed belonged to the component analysis of the study object. Secondly, the main components of the research object play an extremely important role in the production and training of the experimental model. This experiment was conducted based on the main abnormal types of the electrocardiogram of the research object. We have mentioned the background and purpose of this experiment in the introduction module of Title 1, and we believe that the influence of the main components of the investigated population on the experiment can be fully recognized through this module. Thanks again for your advice.

Point 3: In the last sentence in 3.5.3, whether the initial letter is capitalized or not, and in the last sentence in 3.5.2, whether the initial letter is capitalized or not; and, the icons in Figure 2 are inconsistent with other icons in text, and the sizes of the four figures (a), (b), (c), and (d) in Figure 8 are inconsistent with the sizes of the four figures (a), (b), (c), and (d) in Figure 9.

Response 3:Thank you for reminding me. This has played a very important role in improving the quality of our articles. We changed the case of the first letter of the last sentence in subheading 3.5.3 and checked the case of the rest of the article. In addition, Figure 2 has been corrected and reinserted to ensure consistency between text and ICONS in the image, Figure 8 and Figure 9 have been modified and reinserted, and the subscript format and size in the image have been modified to make the subscripts in the image consistent. Finally, thank you again for your professional advice.

Point 4: The materials and methods do not explain how to perform the model evaluation.

Response 4:Thank you for your professional reminder. We have added the following to the 2.4 Modeling module The training set is used to train the model for many times, and then the model coefficients are used to identify and classify the test set. Validation sets were used to verify the model performance, and the model performance was evaluated from the degree of differentiation and calibration.

Point 5: The F1 scores mentioned in the table of evaluation indicators are not covered in the 2.5 statistical methods.

Response 5:Thank you for reminding me. We have added F1 scores to the 2.5 Statistical Methods module, see Line 231 for details.

Point 6: How many doctors diagnosed the ECG physical examination results? If multiple, how to deal with different diagnostic results among doctors.

Response 6:Thank you for your reminding. In our study, we invited three cardiologists with associate senior titles to perform diagnostic electrocardiogram physical examination results. In our study, the diagnostic method was based on the results of three experts' joint diagnosis as the gold standard of electrocardiogram examination, that is, the electrocardiogram results of each worker were respectively diagnosed by three cardiologists with associate senior titles. If the results of three people were consistent, the diagnosis result was the diagnosis result; if the results of three people were inconsistent, the principle of majority rule was adopted. The diagnosis was based on the consultation results. Due to an oversight, we did not add the number of diagnostic doctors and the methods to deal with the different diagnosis results of different doctors into the paper. We are deeply sorry. We plan to add related content to subheading 3.2 to describe the method of selecting the number of doctors and handling different diagnoses among different doctors. The completion of this part will greatly improve the integrity of our research. Sincere thanks again.

Point 7:Some parts of the discussion are repeated with the introduction

Response 7:Thank you for reminding me. We are sorry that some parts of the discussion overlap with the introduction. Due to the limited writing skills, we have reviewed both of them repeatedly, but still have not found the problem you raised. I take the liberty to ask you to point out the problem. Thank you again for your hard work, which is very important to improve our writing level.

Point 8:Reference 23 "Research on automatic recognition algorithm of static ECG based on machine learning" is marked above "python software" in the paper, 24 "Application of Python language in big data analysis "The two are reversed.

Response 8:Thank you for reminding me. Due to our negligence, we marked the two incorrectly, which affected your reading. Now we have corrected the two.

Point 9: The higher incidence of disease in individuals over 60 years of age may be a selection bias. Can the authors explain the solution to this problem?

Response 9: First, we appreciate your valuable input and we agree that there would be a selection bias if workers over 60 were not included. However, the data for this study were obtained from worker occupational health checks conducted on active coal miners, and workers over 60 years of age had retired and no longer received occupational health checks, so they were not included in this study. Based on your professional input, we can then develop a research plan for retired workers and make the research more convincing. We added the following to the discussion section: “However, due to the limitation of data collection, retired workers over 60 years old were not included in the study, which may lead to the deviation of the research results. Follow-up studies can be conducted on retired workers to assess the impact of age on bone mineral density of underground coal mine workers. ‘’

Point 10: The research results of "Ding Hua" cited in the discussion section deviate greatly from the content of the present study, should this citation be revised.

Response 10: As for the discussion part, the topic quoting the research results of Dinghua is the epidemiological survey of electrocardiogram of students in Shenyang Institute of Physical Education. Compared with the electrocardiogram project of coal workers studied by us, firstly, there are differences in the research objects and secondly, there are some differences in the research content. After careful consideration by the team, sports students have high exercise intensity. There is no comparison between the two. The purpose of our discussion is to explain the influence of dust, noise, high temperature and other occupational harmful factors on coal workers in their working environment. In order to further improve the discussion, We cite Wu Yiqin in Yixing textile printing and dyeing enterprise staff health analysis of the point of view for discussion.

Reviewer 2 Report

The authors describe their development of a convolutional neural network (CCN) algorithm to automate interpretation of ECG tracings in 3,392 coal workers in China.  If I am understanding the manuscript correctly, the gold standard used to train the CNN was a single cardiologist with 10 years of clinical experience reading ECGs.  The sensitivity, specificity, and overall diagnostic accuracy of the CNN appears to have been quite high (>90%) for a range of cardiac rhythm abnormalities.  In general, tools like this are certainly needed in clinical medicine that can assist busy clinicians by providing automated interpretation of clinical data to detect pathology quickly and accurately.

This manuscript is promising, but would benefit from significant revision as follows:

1)    The manuscript is full of typos, grammatical errors, and duplicate statements (see examples below).  The article should be review carefully by a fluent English speaker and revised to conform to standard English grammar.

2)    It is not crystal clear to me what the gold standard was: was it the single cardiologist with 10 years of experience reading ECGs or something else?  If the single cardiologist, why not have a second cardiologist read a subset of these ECGs to determine the inter-rater reliability of the “gold standard”?

3)    Can the authors comment on the speed of implementation of this algorithm?  I.e., you mention that your CNN algorithm is more accurate than existing ones.  Does that come at a cost of a higher number of computations?  If so, does that have any potential to limit the clinical usability of your algorithm as a real-time monitoring tool?

Further, I’m a physician with only a superficial awareness of machine learning, so I am unable to comment on the machine learning methodology.  I would ask the Editor to select at least one reviewer with expertise in CNN to comment on this aspect of the manuscript.

Representative (but not complete) list of grammatical and typographical errors:

Lines 35-43: Long sentence with several problems: (1) many clauses connected by "ands" when they should be connected by commas or semi-colons; (2) likely a typo in the middle of the sentence because the sentence is not understandable ("... but due to the large worker The large base of physical examiners ...").

Line 46: "thus misdiagnosis and affect the accuracy".  These 2 things ("misdiagnosis" and "affect accuracy") are essentially the same thing stated twice.

lines 64-65: "their model yielded 64 results expert compared to expert” - please review and revise this for grammatical coherence

lines 80-83: please have sentence reviewed by a fluent English speaker

line 86: “In this study, a proximate study was conducted” – unclear what authors mean by a “proximate study”

Lines 92-94: the same sentence is repeated twice.

Author Response

Point 1: The manuscript is full of typos, grammatical errors, and duplicate statements (see examples below).  The article should be review carefully by a fluent English speaker and revised to conform to standard English grammar.

Response 1:Thank you for your valuable advice. We are very sorry for the problems you raised. We have transferred the article to professional English speakers in our team for modification and adjustment. See Point 4-Point 9 for example modifications. Thank you again for all your hard work.

Point 2:  It is not crystal clear to me what the gold standard was: was it the single cardiologist with 10 years of experience reading ECGs or something else?  If the single cardiologist, why not have a second cardiologist read a subset of these ECGs to determine the inter-rater reliability of the “gold standard”?

Response 2:Thank you for your reminding. In our study, we invited three cardiologists with associate senior titles to perform diagnostic electrocardiogram physical examination results. In our study, the diagnostic method was based on the results of three experts' joint diagnosis as the gold standard of electrocardiogram examination, that is, the electrocardiogram results of each worker were respectively diagnosed by three cardiologists with associate senior titles. If the results of three people were consistent, the diagnosis result was the diagnosis result; if the results of three people were inconsistent, the principle of majority rule was adopted. The diagnosis was based on the consultation results. Due to an oversight, we did not add the number of diagnostic doctors and the methods to deal with the different diagnosis results of different doctors into the paper. We are deeply sorry. We plan to add related content to subheading 3.2 to describe the method of selecting the number of doctors and handling different diagnoses among different doctors. The completion of this part will greatly improve the integrity of our research. Sincere thanks again.

Point 3: Can the authors comment on the speed of implementation of this algorithm?  I.e., you mention that your CNN algorithm is more accurate than existing ones.  Does that come at a cost of a higher number of computations?  If so, does that have any potential to limit the clinical usability of your algorithm as a real-time monitoring tool?

Response 3:First of all, thank you very much for your professional questions, which are very helpful for us to improve the quality of the paper. Because there are too many variables describing the time complexity of deep learning, it cannot be accurately described. However, sorting algorithms can determine the time complexity as long as they know the total amount of data, which is a free variable. And we can also measure the calculation speed of the algorithm through the index of time complexity. We add the following in lines 253 to 260, “

: the depth of the network.

C1: number of convolutional kernels in this layer.

For the L-th convolution layer, the number of input channels  is the number of output channels for the L-1st convolution layer. The time complexity of this model is:

According to the number of parameters, we can know that the accuracy of the algorithm is high mainly because the number of parameters is relatively large, and the accuracy is increased by many limiting factors. In addition, high accuracy is closely related to the establishment and structure of the model, but not necessarily related to the number of calculations and calculation speed. For the clinical availability you proposed, we use Kappa value, positive predictive value and negative predictive value to measure, evaluate the four typical diseases respectively, and add their values to Table 7-9. As can be seen from the data in the table, the results of the three are close to the ideal value of 1 (100%).

Point 4:Lines 35-43: Long sentence with several problems: (1) many clauses connected by "ands" when they should be connected by commas or semi-colons; (2) likely a typo in the middle of the sentence because the sentence is not understandable ("... but due to the large worker The large base of physical examiners ...").。

Response 4:Thank you for reminding me. We have modified the long sentences, making adjustments for spelling, grammar, word collocation and so on. Due to the excessive content involved, it is impossible to elaborate in the reply. Please refer to the manuscript for details.

Point 5: Line 46: "thus misdiagnosis and affect the accuracy".  These 2 things ("misdiagnosis" and "affect accuracy") are essentially the same thing stated twice.

Response 5:Thank you for reminding me. The accuracy here refers to the accuracy of diagnosis. We have added and modified the sentence of the article in line 46.

Point 6: lines 64-65: "their model yielded 64 results expert compared to expert” - please review and revise this for grammatical coherence

Response 6:64 is not the number of results, it's line 64. We have modified the original sentence to” Hannun A Y et al. 's machine learning group used a 34-layer convolutional neural network to detect arrhythmias. Compared with experts, the AUC value of their model reached 0.97, and the sensitivity was 0.83, which was higher than the expert level of 0.78.”

Point 7: lines 80-83: please have sentence reviewed by a fluent English speaker.

Response 7:Thank you for your reminding. We have changed the sentence in line 80-83” The purpose of this study is to assist doctors in electrocardiogram identification, reduce the workload of electrocardiogram diagnosis, and control the error of manual diagnosis. It provides a reference for the physical examination electrocardiogram diagnosis of occupational population and also provides help for the prevention and treatment of cardiovascular diseases.”

Point 8: line 86: “In this study, a proximate study was conducted” – unclear what authors mean by a “proximate study”

Response 8:Thank you for your reminding. Due to translation, the wording is inappropriate. We revised the original sentence.” the current situation study method was used in this study”.

Point 9: Lines 92-94: the same sentence is repeated twice.

Response 9:Thank you for reminding us that we have removed the duplicate statements.

Round 2

Reviewer 2 Report

The manuscript continues to be full of grammatical errors/inconsistencies.  Every word of this manuscript needs to be reviewed and revised by a fluent English speaker.

I will give just a few examples of statements that are strange or grammatically problematic.

line 34: "Electrocardiogram,which detects electrophysiological signals from the heart muscle, is  considered a perfect monitoring method."  

What is a "perfect monitoring method"? The word "electrocardiogram" should have a "The" in front of it.

line 37: "ECG has great reference value for clinical diagnostic studies."

The meaning of this statement is not clear to an English speaker.

lines 39-42: "Reliable automated interpretation of ECG 39 signals is extremely beneficial to clinical routine and patient safety. However, due to 40 the large number of physical examination personnel and the heavy workload, there 41 are inevitably omisses in the recognition and diagnosis of electrocardiogram by doctors." 

The second sentence contains the word "omisses," which as far as I'm aware is not used in the English language (I assume you meant omissions).  And the 2 sentences should be presented in the reverse order to make logical sense.

Author Response

Point 1:line 34 "Electrocardiogram,which detects electrophysiological signals from the heart muscle, is considered a perfect monitoring method."What is a "perfect monitoring method"? The word "electrocardiogram" should have a "The" in front of it.

Response 1:Thank you for your professional advice. As for the problem you raised, we have asked professionals to revise it.

Point 2:line 37 "ECG has great reference value for clinical diagnostic studies."The meaning of this statement is not clear to an English speaker.

Response 2:First of all, thank you for your question. We are very sorry that some sentences are impassable due to our limited English proficiency. For the above problems, we have asked professional staff to modify it.

Point 3:lines 39-42 "Reliable automated interpretation of ECG 39 signals is extremely beneficial to clinical routine and patient safety. However, due to 40 the large number of physical examination personnel and the heavy workload, there 41 are inevitably omisses in the recognition and diagnosis of electrocardiogram by doctors."The second sentence contains the word "omisses," which as far as I'm aware is not used in the English language (I assume you meant omissions). And the 2 sentences should be presented in the reverse order to make logical sense.

Response 3:Thanks for your professional advice, we have asked professionals to modify it. Thanks again for your advice.
